# Description and Analysis of Research on Death and Dying during the COVID-19 Pandemic, Published in Nursing Journals Indexed in SCOPUS

Leticia Cuellar-Pompa [1],*, José Ángel Rodríguez-Gómez [2], María Mercedes Novo-Muñoz [2], Natalia Rodríguez-Novo [2], Yurena M. Rodríguez-Novo [3] and Carlos-Enrique Martínez-Alberto [4]

1. Instituto de Investigación en Cuidados del Ilustre Colegio de Enfermeros de Santa Cruz de Tenerife, Calle San Martín, 63, 38001 Santa Cruz de Tenerife, Spain
2. Departamento de Enfermería, Facultad de Ciencias de la Salud, Sección de Enfermería y Fisioterapia, Universidad de La Laguna, Sta. María Soledad, s/n, Apartado 456, C. P., 38200 San Cristóbal de La Laguna, Spain; jarogo@ull.edu.es (J.Á.R.-G.); mernov@ull.edu.es (M.M.N.-M.); nrodrigu@ull.edu.es (N.R.-N.)
3. Hospital Universitario Nuestra Señora de la Candelaria, Carretera General del Rosario, 145, 38010 Santa Cruz de Tenerife, Spain; yrodrign@ull.edu.es
4. Escuela de Enfermería Nuestra Señora de Candelaria, Carretera General del Rosario, 145, 38010 Santa Cruz de Tenerife, Spain; cmaralbp@gobiernodecanarias.org
* Correspondence: lcuellarpompa@iic-enfermeriacanaria.com

**Abstract:** Aim: To offer an overall picture of the research published regarding the different aspects of death and dying during the COVID-19 pandemic in journals covering the field of nursing in the Scopus database. Design: bibliometric analysis. Methods: The metadata obtained were exported from Scopus for subsequent analysis through Bibliometrix. Using the VOSviewer co-word analysis function, the conceptual and thematic structure of the publications was identified. Results: A total of 119 papers were retrieved, with the participation of 527 authors. The publications were found in 71 journals covering the nursing area. The main lines of research revolved around the keywords "palliative care" and "end-of-life care" in regard to the ethical, psychological, and organizational challenges faced by the health professionals who cared for these patients. Conclusion: The results obtained offer a range of data and images that characterize the scientific production published on this topic, coming to the conclusion that, due to the multifaceted and multidisciplinary approach to the experience of death, care, and accompaniment in the dying process, bibliometric maps improve the comprehensive understanding of the semantic and conceptual structure of this field of research. This study was retrospectively registered with the OSF Registries on the 14 March 2024.

**Keywords:** nurses; palliative care; attitude to death; death; bibliometrics; knowledge discovery

## 1. Introduction

Bibliometrics, as a discipline, allows for the description and evaluation of scientific publications in any field of knowledge [1]. This makes it possible to define the state of research, as well as analyze the different characteristics of scientific activity and predict trends through the analysis of different indicators that are established from the numerical data extracted from academic and scientific publications [2], such as the frequency of publication, authorship, collaboration, citation or co-citation by countries, institutions, etc. Thus, it is also possible to analyze the thematic development or the conceptual structure of a field through the analysis of keywords. Based on these considerations, this study focused on the analysis of publications on death and dying during the COVID-19 pandemic that were published in nursing journals indexed in Scopus.

The health crisis caused by the pandemic represented a great challenge for health systems globally due to the drastic consequences caused by the sudden hospital overload. This scenario increased the pressure on healthcare professionals, with the workload resulting in

increased stress, physical and emotional exhaustion, fear, loneliness, the risk of contagion and death [3–5], which led to, according to data from 44 countries, the International Council of Nursing announcing at the end of 2020 that as many nurses had died from COVID-19 as during the First World War [6].

The health professionals had to dedicate themselves full time to their care activities. In this context, other common activities, including research, were relegated to the background [7]. This was made clear in a study in which the authors found a marked increase in nursing publications related to COVID-19 in the first months of 2021 compared to what was published during 2020 when the worst stage of the pandemic occurred [7].

Different bibliometric studies, where the keywords of the publications on nursing production throughout the pandemic were analyzed, agree that, in the first stage, the research was related to topics on coronavirus infection, infection control and health and nursing policy, after which the focus changed to the effects of the pandemic on mental health and quality of life, as well as issues related to nursing education, types of online training, nursing students, and new experiences of remote assistance [8–10].

Although death is part of the life cycle [11], the data on the impact experienced by health professionals and the large number of deaths associated with COVID-19 were frequently discussed in the media [12,13]; however, the different aspects of the process of death and dying during this period do not seem to have been developed in the scientific literature. Furthermore, there do not seem to be any bibliometric studies addressing this issue, although there are several bibliometric publications on COVID-19 in relation to nursing [10,14].

Nurses must provide care to both patients and family members, and they have to deal with death and the fear of facing this [15]. On the other hand, due to the relevance of bibliometric research in identifying, understanding and promoting problem solving [16], this study aims to provide a more precise picture of the quantity and characteristics of international scientific production published in the journals covering the field of nursing in the Scopus database, examining all issues related to the processes of death and dying.

## 2. Materials and Methods

There are two methodological categories for the development of bibliometric studies [17], which are scientific performance and scientific mapping. Performance analysis is, by its very nature, descriptive. It has been established as the distinguishing feature of bibliometrics analyses because it concentrates on the contributions made by the contributors (author, journal, institution, country, etc.) of a particular field of knowledge within a given discipline [18]. The maps of science, focus on the analysis of the connections between these contributions in terms of the analysis of annual production, citations, co-citations, co-words, or co-authorship. The combination of these analyses with network analysis makes bibliometrics an indispensable tool for presenting the bibliographic and intellectual structure of a research area [19,20].

### 2.1. Aims

- To develop a statistical-descriptive analysis of the publications retrieved with the search.
- To analyze the main indicators using Bibliometrix 4.1.4.
- To visualize the main concepts and focuses of interest in the research based on the analysis of co-words in VOSviewer 1.6.16.

### 2.2. Study Desing

This is an exploratory, descriptive, longitudinal, and retrospective study about the different aspects of the process of death and dying during the COVID-19 pandemic. It was conducted through two bibliometric tools to analyze the publications of the journals of nursing indexed in the Scopus database. Ethical approval was not required for this

study as it did not involve the use of data from human participants, relying solely on bibliographic data.

*2.3. Information Sources*

The Scopus database was selected, as it is a fundamental source of information for the evaluation of the research and also contains all the bibliographic data available. The records were downloaded in May 2023.

Firstly, a search strategy was designed to obtain the corpus of information using the following fields: title, abstract and keywords. The search was limited to the years 2020–2023 and the subject area "Nursing". The search strategy and the results are shown in Table 1.

**Table 1.** Search strategy performed in Scopus on 9th May 2023.

| No. | Query | Results |
|---|---|---|
| #5 | (LIMIT-TO ((PUBYEAR, 2023) OR LIMIT-TO (PUBYEAR, 2022) OR LIMIT-TO (PUBYEAR, 2021) OR LIMIT-TO (PUBYEAR, 2020)) AND LIMIT-TO (SUBJAREA, "NURS")) | 119 |
| 4 | #1 AND #2 AND #3 | 665 |
| 3 | TITLE-ABS-KEY (grief OR bereavement OR mourning OR "End of life care" OR end-of-life OR "end of life" OR "End of life decision-making" OR suicide OR "Traumatic Death" OR death-related OR "Death and Dying" OR "Attitude to death" OR "Death education") | 253,309 |
| 2 | TITLE-ABS-KEY (thanatology OR religion OR religious OR spiritual OR spirituality OR philosophy OR anthropology OR sociology OR socialization OR cultural OR ethical OR legal OR institutional OR "Life Span" OR "Legal Aspects" OR "Historical Perspectives" OR "Contemporary Perspectives" OR "Professional Issues") | 2,695,503 |
| 1 | TITLE-ABS-KEY (pandemic OR covid-19 OR sars-cov-2 OR "Severe Acute Respiratory Syndrome Coronavirus 2" OR "NCOV" OR "2019 NCOV" OR 2019-ncov OR "Novel Coronavirus" OR "Coronavirus disease 2019") | 570,295 |

*2.4. Bibliometric Data Analysis*

The analysis parameters included the data corresponding to production by countries, authors, journals, and institutions. In addition, the main concepts and focuses of interest were identified through the co-occurrence analysis of the indexed keywords (IKs), which are terms chosen by Scopus and standardized based on the vocabulary of the Elsevier thesaurus [21].

The metadata obtained were exported from Scopus, in txt format, for subsequent analysis through Bibliometrix, an open-source program, developed in R, which provides a set of tools for the scientometric analysis of the literature. This program allows for the analysis of various bibliometric indicators within a corpus of bibliographic information [22,23].

In the same way, using the VOSviewer co-word analysis function [24,25], the conceptual and thematic structure of the publications was identified by means of the graphic representation of the different lines of research [26,27]. Labeled and density maps were chosen from the types of representations available. The former uses labels with the names of each term, and the sizes of these labels are proportional to their weight (frequency). Density maps are characterized by the representation of nodes in colors ranging from blue to green and yellow to red. These colors reflect the density of the relationships between the terms (co-occurrence). When the density or co-occurrence is higher, the hue on the map will be closer to red; in contrast, with greater dispersion and, therefore, less co-occurrence, the hue on the map will turn bluer [25].

VOSviewer combines visualization and clustering techniques to favor the analysis; thus, for the positioning of the IKs on the map, the VOS (visualization of similarities) technique was used [28], which builds a similarity matrix from the matrix of co-occurrence, using the Association Strength measure as a similarity index to normalize the bibliometric network [29]. The Association Strength is based on the normalization of co-occurrence values and is applied so that these values adequately represent the analyzed corpus and facilitate clustering.

As a phase prior to the analysis in VOSviewer, the file with the keywords was subjected to a normalization process with the aim of cleaning the information collected to improve its quality. In this process, the duplicate elements and those that represented the same concept (but were written differently) were unified, while those that contained errors in writing were corrected. A thesaurus was then created with this data, which was then uploaded to VOSviewer to promote clarity in the representation of the maps. Likewise, some terms used in the search strategy (COVID-19, coronavirus, pandemic, etc.) were dispensed with, as well as those terms referring to types of publication (article or clinical article, etc.) and the study design (review, qualitative research, surveys, questionnaires, etc.). Similarly, high-frequency IKs with zero input into the representation, such as the keywords human or humans, were removed from the analysis. Finally, in this case, due to low frequency, the names of countries (USA, Canada) were also eliminated.

Several trials were carried out where different frequency thresholds were tested until it was verified that the map with the best visualization criteria was obtained by limiting the frequency parameter to $\geq 4$ appearances. Thus, it was possible to distinguish the thematic structure of the analyzed research more clearly based on the eighty-two most frequent terms.

A clustering algorithm was executed [30], using the VOS technique, that positions and classifies the keywords to be mapped. This algorithm allows for different resolution parameters depending on the value provided for their configuration; thus, a low resolution parameter simultaneously implies a decrease in the number of generated clusters and vice versa. After several tests applying different resolution values, and once the different test results were analyzed, it was decided that the resolution parameter should be kept with a value of 1, which is configured by default in VOSviewer. The minimum size of the clusters was configured to a value $\geq 3$ IK, guaranteeing a minimum of consistency in the thematic groups.

## 3. Results

### 3.1. Descriptive Analysis

A total of 119 [31–149] works were retrieved by the search strategy in Scopus. The complete metadata, exported from Scopus, is available in the supplementary material attached to this publication (File S1) [31–149]. The data obtained showed that these documents were published in 71 journals in the nursing area, featuring the participation of 527 authors. An amount of 549 signatures were counted, which comes to an average of 4.43 authors per document. There were 89 registered articles and 14 reviews in addition to less-frequent documentary typology. The United States and the United Kingdom were, respectively, the countries with the highest number of publications. Table 2 shows an overview of the main results of the analysis.

The most productive year was 2021, with 45 publications. In 2020, 19 works were published, and 16 references have been registered to date (9 May) in Scopus in 2023 related to the subject of study.

**Table 2.** Main Information about the collection, from Biblioshiny for Bibliometrix.

| Description | Results |
|---|---|
| **Main Information about Data** | |
| Timespan | 2020:2023 |
| Sources (Journals, Books, etc.) | 71 |
| Documents | 119 |
| Average years from publication | 1.6 |
| Average citations per documents | 7.4 |
| Average citations per year per doc | 2.3 |
| References | 4191 |
| **Document Types** | |
| article | 89 |
| book | 2 |
| book chapter | 3 |
| editorial | 6 |
| letter | 2 |
| note | 3 |
| review | 14 |
| **Document Contents** | |
| Keywords Plus (ID) | 750 |
| Author's Keywords (DE) | 350 |
| **Authors** | |
| Authors | 527 |
| Author Appearances | 549 |
| Authors of single-authored documents | 12 |
| Authors of multi-authored documents | 515 |
| **Authors Collaboration** | |
| Single-authored documents | 12 |
| Documents per Author | 0.23 |
| Authors per Document | 4.43 |
| Co-Authors per Documents | 4.61 |
| International co-authorships (%) | 22.7 |
| Collaboration Index | 4.81 |

*3.2. Journals Analysis*

Bradford's law is a useful tool for analyzing the distribution of information, as well as to better understand its structure and how the different aspects of a topic are related to one another. Taking this into account, the present analysis established that, of the 71 journals reported, the Journal of Pain and Symptom Management (Q1) with 11 publications [57,62,65,70,77,81,87,99,103,109,124], the Journal of Palliative Medicine (Q2) with 10 [59,71,89,91,93,96,98,131,148], the Journal of Religion and Health (not JCR) with 7 [58,63,67,78,107,112,118], Nursing Ethics (Q1) with 6 [39,56,69,73,75,104], Journal of Medical Ethics (Q1) with 4 [53,100,102,106] and BMJ Supportive and Palliative Care (not JCR) with 3 publications [68,85,97] were the central sources, with accounting for only 8.5% of the total. See Figure 1.

All the above-mentioned journals are on the thematic area of nursing, which includes general nursing; problems, ethics and legal aspects; medical and surgical nursing; and oncology nursing, gathered from the Scopus database.

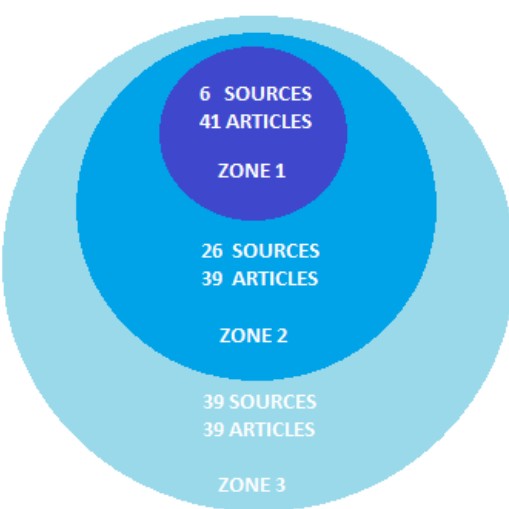

**Figure 1.** Bradford's Law, applied to articles on death and dying published in the Scopus nursing journals between 2020 and 2023.

The theme addressed in the papers published by these six central journals revolved around end-of-life care, palliative care, patient dignity, and the ethical challenges in critical care and end-of-life care during the pandemic. In addition, support for grief, death anxiety, mental health in relation to stress, as well as the moral conflicts experienced by nurses in the midst of the situation generated by the health crisis, were also addressed. Several studies have been developed on the perspective of nursing students and medical residents in regard to death and dying and the ethical challenges related to end-of-life care. Other issues developed had to do with spirituality and religion as coping strategies in the midst of the health emergency situation.

### 3.3. Authorship and Productivity Analysis

The most productive authors were Betty R. Ferrell and William E. Rosa, each with three co-authored publications [51,114,124]. Two papers were published by 3.4% of the authors, while 96.2% of the authors had only one publication. These results correspond to the premise of Lotka's Law regarding the unequal distribution of productivity. This law states that a few authors publish most of the papers on a research topic, while the majority publish a minimum number of studies. See Table 3.

**Table 3.** Author Productivity Using Lotka's Law.

| Documents Written | No. of Authors | Proportion of Authors |
|:---:|:---:|:---:|
| 1 | 507 | 0.962 |
| 2 | 18 | 0.034 |
| 3 | 2 | 0.004 |

### 3.4. Scientific Production by Countries

Only 10.5% (United States, Canada, Spain, and United Kingdom) out of a total of 38 countries published approximately 49% of the works. See Figure 2.

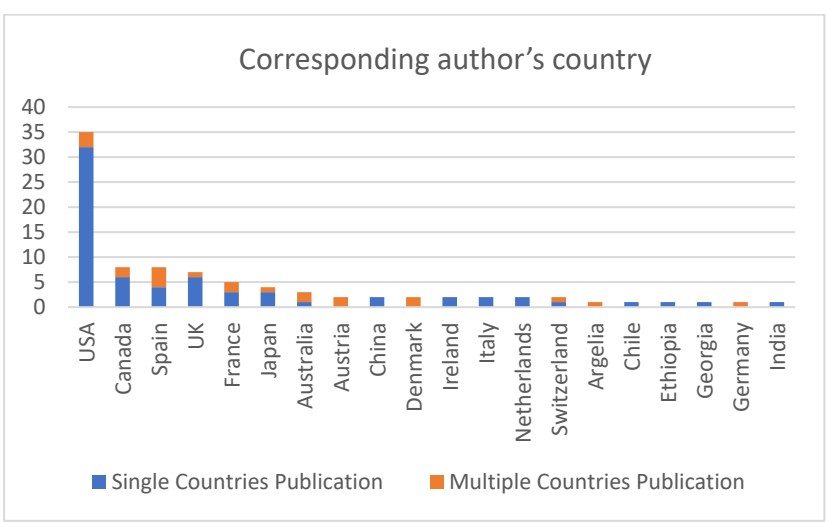

**Figure 2.** Distribution of publications according to the country of the corresponding author.

*3.5. Institutional Scientific Production*

Bibliometrix performs a disambiguation task with the goal of cleaning the data to account for single institutions. The ambiguity in this case refers to the different ways of representing the institutional names in the bibliographic metadata, which can involve misspellings or the writing of acronyms or abbreviations of the name, which makes them seem different when they refer to the same institution.

Of the 234 institutions, the 10 most relevant were selected (see Table 4). Seven were universities, while the rest were medical centers or hospitals.

**Table 4.** Frequency of publication of the 10 most productive institutions in the analyzed period.

| Affiliation | Freq. |
| --- | --- |
| University of California, San Diego | 12 |
| Massachusetts General Hospital | 10 |
| University of Toronto | 10 |
| King's College, London | 9 |
| Maastricht University Medical Center | 9 |
| Mayo Clinic | 9 |
| Maastricht University | 8 |
| Icahn School of Medicine at Mount Sinai | 7 |
| The Ohio State University | 7 |
| University of Michigan Medical School | 7 |

*3.6. Main Topics*

Of the 119 documents recovered, 14.3% [17] did not have IKs, so they were not included in the analysis.

After applying the clustering algorithm, with a resolution parameter of one, five clusters were generated that reflected the degree of similarity between the IKs. The clusters were visualized by a labeled bibliometric map and a bibliometric density map.

The size of the clusters in the labeled bibliometric map was determined by different factors, such as the number of IKs within each group, as well as the frequency and the weight or similarity index. The color of each cluster was randomly assigned by the program (see Figure 3). The five clusters represented on the map defined the main topics published in Scopus nursing journals in the defined period and on the subjects related to death and dying in times of pandemic.

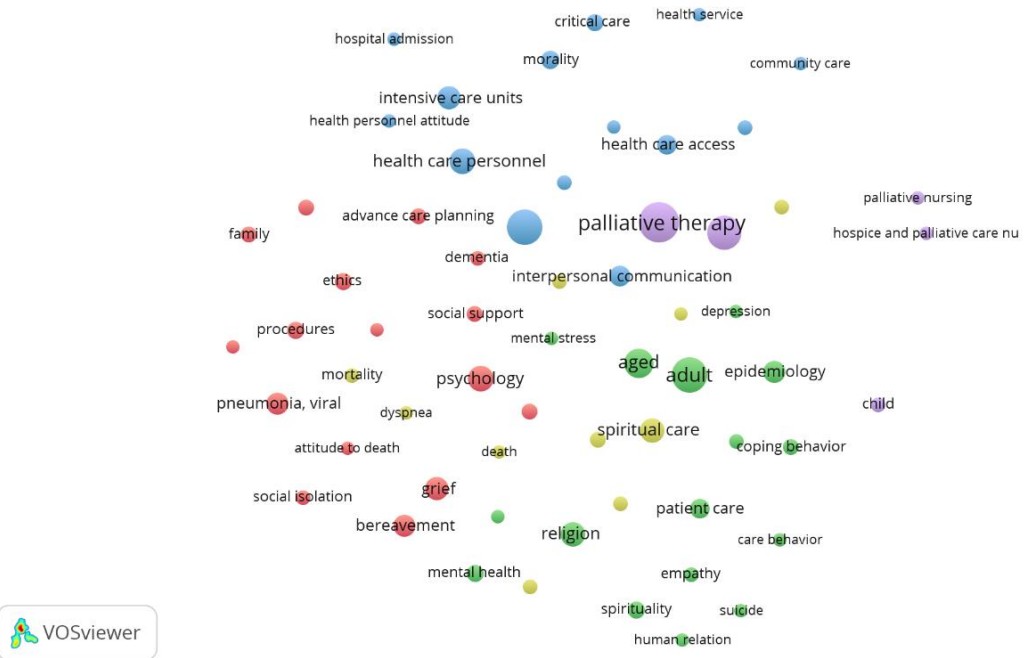

**Figure 3.** Labeled bibliometric map where the 82 analyzed IKs are represented. The labels and the size of the circles reflect the weight of the IK, while the color of the circles shows the thematic group which each IK belongs to. VOSviewer assigns the color of the clusters automatically by applying the VOS algorithm, as explained in the methods section.

The following section provides a detailed description of the contents of each cluster. Table 5 displays the most significant IKs within each cluster, along with their frequency of appearance and corresponding thematic group color.

**Table 5.** IKs with the greatest weight within each cluster (the color corresponds to the different identified thematic groups).

| Cluster | Color | Indexed Keywords | Occurrences |
|---|---|---|---|
| 1 | | Psychology | 15 |
| 1 | | Grief | 12 |
| 1 | | Bereavement | 11 |
| 1 | | Pneumonia, viral | 11 |
| 2 | | Adult | 28 |
| 2 | | Aged | 19 |
| 2 | | Religion | 14 |
| 2 | | Epidemiology | 11 |
| 3 | | End-of-life care | 28 |
| 3 | | Health care personnel | 15 |
| 3 | | Intensive care units | 12 |
| 3 | | Interpersonal communication | 10 |
| 4 | | Spiritual care | 13 |
| 4 | | Personal experience | 6 |
| 4 | | Mortality | 5 |
| 4 | | Dying | 5 |
| 5 | | Palliative therapy | 35 |
| 5 | | Palliative care | 26 |
| 5 | | Child | 5 |
| 5 | | Palliative nursing | 4 |

Cluster 1 (red): in general, publications were found on family, social, and healthcare support in the face of loss and mourning, as well as ethical, cultural, and psychological factors related to attitudes towards death and advanced care planning.

Cluster 2 (green): publications on spirituality and the religious perspective in the face of anxiety, anguish, depression, and stress, as well as the role of empathy, human relations, and coping behaviors in the mental health of adults and the elderly, in addition to the development of care by health personnel and the problematic issue of suicide in nurses.

Cluster 3 (blue): this covers the organizational and administrative point of view regarding care for the critically ill and those at end of life in intensive care units from the perspective of health personnel, particularly nurses, in relation to policies, procedures, health care systems and services, as well as access to therapeutic measures to alleviate suffering in critical patients when faced with the challenge of communication with the patient and the family in the midst of confinement and strict isolation measures during the pandemic.

Cluster 4 (yellow): Publications on death and dying from a clinical, spiritual and psychosocial perspective. The problematic issue of the well-being of patients in regard to the control of symptoms and complications derived from COVID-19, as well as the situation of loneliness and isolation during the dying process in the period of the pandemic; the personal experiences of both patients and their families regarding the medical care received; spiritual care in this environment, the importance of training nursing staff in spiritual care, and the resources and barriers to providing spiritual care or experiences with spirituality in clinical practice were some of the issues addressed, in addition to the psychosocial perspective of care in the midst of mobility and accompaniment of dying patients' restrictions.

Cluster 5 (purple): IKs related to different aspects of palliative care were included in this cluster.

Different nuclei related to the most developed themes were identified in the bibliometric density map (see Figure 4). The fields with the greatest IK impact were identified in the center of the map (in red), and several foci of interest were found. These central groups are connected to one another by different bridge nodes, as well as with the rest of the groups on the periphery of the map, which present less interaction (in yellow, green, and blue).

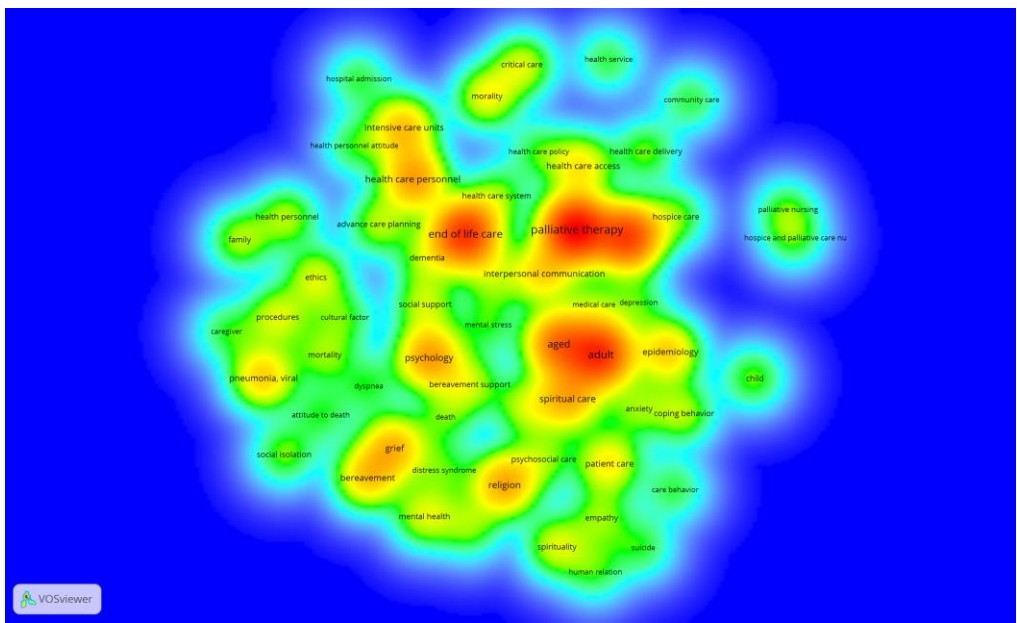

**Figure 4.** Bibliometric density map showing the 82 analyzed IKs. The colors close to red show the areas with the highest density of IKs co-occurrences; the colors close to yellow and green show the areas with the lowest density IKs co-ocurrences.

The prevailing lines of research were developed around palliative and end-of-life care in adult patients from the perspective of health personnel in intensive care units. There was also published material about health care systems and policies, as well as access to them, the provision of health care to the elderly with dementia in hospices and residences, and the evident need, due to the health collapse as a result of the pandemic, of preparing advance care plans on palliative care for out-of-hospital patients.

The lower peripheral zone of the map shows the main interest in psychological or mental health issues surrounding the dying process regarding loss and mourning, anxiety, and anguish due to fear of contagion and death. In this respect, works were developed that addressed the importance of religion and spirituality as a coping strategy and support for loss and mourning.

The peripheral zone located on the left refers to studies related to attitudes towards death and the ethical conflicts experienced by nurses faced with the challenge of having to cope with the care of patients with COVID-19 in the midst of the health collapse, with high mortality rates, aggravated by mandatory social isolation and loneliness and suffering not only experienced by patients but also by family members and caregivers who, in some cases, were forced to prioritize attention and care.

There are two apparently isolated nuclei on the right which are related to pediatric palliative care in patients with other pathologies, such as cancer.

## 4. Discussion

The current study explored the process of death and dying in nursing journals from the Scopus database using bibliometric techniques such as quantitative and visual analysis.

According to the outcomes of this study, there seems to be a pattern in terms of the increase in publications in the midst of the different public health crises of recent years. This was clearly portrayed in a bibliometric study, which described the trends in global research and the characteristics of the publications on the different emerging pathogens, which was included in the Blueprint list of priority diseases of the World Health Organization (WHO). In this work, the authors found that the growth of publications had experienced a significant peak coinciding with the years when an outbreak was declared. Thus, there was a large increase in publications on SARS, MERS, and other emerging diseases, such as the Ebola virus, during the outbreaks that occurred in 2003, 2012, and 2014, respectively [150].

The year 2020 was an unprecedented milestone in research due to the high number of publications on COVID-19. In contrast, the results of the present study suggest that approximately 71% of the papers were published between 2021 and 2022, coinciding with the end of the critical period of the pandemic, although there was a decrease of 7% in 2022 compared to the previous year, which has since increased up to the present. These results agree with those of other authors, who found an almost threefold increase in nursing publications on COVID-19, starting in 2021, compared to 2020 [151]. This difference between publications on COVID-19 in general, and nursing publications in particular [152–154], is probably associated with the intense care activity that took place in hospitals and health centers during the course of the pandemic, which could have limited nursing scientific productivity during 2020. On the other hand, it should be remembered that the present analysis corresponds to a highly specific topic in the vast range and number of publications on COVID-19.

Regarding the journals, the results of the present study coincide with previous works where it was concluded that the journals with the highest number of publications (Bradford zone 1) belonged to the highest quartiles in the analyzed category (Q1/Q2 of the JCR) [153,155]. Among the most relevant journals identified by these authors are, as in the present work, the Journal of Pain and Symptom Management and the Journal of Palliative Medicine. Of the six journals, four belonged to publishers from the UK and two from the USA, countries that, in turn, are among the most productive.

The Journal of Pain and Symptom Management was the most productive in the group of main resources. Its publications are on clinical research and good practice related to

alleviating the burden of disease among patients with serious or life-threatening illnesses. In this case, there were articles on end-of-life care, palliative care, critical care planning, euthanasia and bereavement support.

The area of greatest thematic relevance can be identified and compared with other similar studies by applying Bradford's law. Distribution analysis also allows a better understanding of the structure of the information and how different topics are related. This can be especially useful in academia to identify emerging research areas and establish new lines of research.

In general, the theme of the main journals found in the present study (hospice and end-of-life care) correlates with the results of the co-word analysis that will be discussed below.

The short period of time that elapsed since the start of the pandemic and the high interest in publishing could explain the high rate of transience (authors of a single publication). This result is compatible with previous publications which analyzed the scientific production on COVID-19 [156,157].

Only four of the thirty-eight recorded countries were responsible for almost half of the scientific production on the subject. The little international collaboration found is not equivalent to the data from previous studies [151]. However, perhaps due to the thematic specificity of this work, in regard to the volume of results, there is a correlation with previous studies regarding the fact that the countries with the highest productivity also possibly had a higher incidence of infections and deaths from COVID-19 [158].

The prevalence of educational centers among the most productive institutions could be related to the high demand for work in hospitals. Eighty percent of the most productive institutions came from the USA, UK, and Canada. The remaining 20% were located in the Netherlands, specifically Maastricht University and Maastricht University Medical Center+. These two institutions alone published 12.1% of the production of the most productive institutions, which suggests that the contributions of these institutions on this topic should be taken into account.

Despite the fact that China has been at the forefront of world scientific production on COVID-19 [151,159], it should be mentioned that, in the present analysis, China is not among the ten most productive countries, institutions, or authors.

The co-word analysis shows that nursing publications develop various dimensions on the phenomenon of death and dying in the context of the pandemic, both for patients and their families and for health personnel, especially nurses. There is a concern to address the emotional, spiritual, ethical and communication aspects that are affected by the exceptional conditions imposed by COVID-19. Thus, several lines of research focused on mental health and the psychological state of health workers were found. This theme was also developed in previous studies, which demonstrates its relevance [160,161].

In this sense, the bibliographic findings show that nurses bore a very high psychological burden, which coincides with a previous study that examined mental health research during the COVID-19 pandemic [162]. There is also evidence of the need to improve health care systems and services to guarantee dignified and humanized care for critically ill patients and those at the end of life.

The analyzed bibliometric maps show that the fields with the greatest impact of IKs were related to palliative care and end-of-life care in adult patients, from the perspective of health personnel, in intensive care units. These keywords, along with the term "advance care planning", were also the most active in a study that analyzed, using a bibliometric analysis, global research on end-of-life care [155].

The COVID-19 pandemic claimed many lives around the world, presenting unique challenges for the whole of society, especially for health systems and health professionals working with critical or terminal patients and in palliative care, revealing a pressing need for preparation and education for death [163].

The findings of the present study are consistent with those of a group of researchers from universities in Hong Kong, Palestine and Turkey who referred to the relevance of palliative and end-of-life care in the context of the pandemic, as well as the ethical,

psychological, and organizational challenges faced by healthcare professionals caring for these patients [158].

Other studies analyzed here addressed the issue of pediatric palliative care in patients with other pathologies, such as cancer. This line of research appeared as an isolated cluster in the density map and with little interaction with the rest of the IKs, which suggests the prioritization of studies on COVID-19. This fact also coincides with the study cited above, in which the authors identified the need for more research on bereavement and bereavement support for families, health professionals and patients with other life-threatening diseases, such as cancer [158].

The lines of research described reflect the diversity and complexity of the needs and demands of terminally ill patients and their families, as well as the need for comprehensive and interdisciplinary care that addresses all physical, emotional, social, and spiritual aspects in regard to the death and dying process.

The quantitative results are consistent with the content analysis and the main lines of research visible in the two-dimensional bibliometric maps of VOSviewer. The situation regarding published research is consistent with the unique challenges for the health care system and especially for hospice nurses. Nurses were at the forefront of this crisis, accounting for more than half of the workforce of the global medical care profession, having the highest proportion of direct patient care time compared to any other health professional [164], which is why the global outbreak of COVID-19 has been considered a serious risk for health care providers, especially for nurses [165].

*Limitations*

Bibliometric analysis can provide valuable information on scientific production; however, it also has important limitations. One of the main limitations is that it does not take into account the quality of the published papers.

Of the 119 references, seventeen did not contain IKs. This information should be kept in mind when interpreting the results, since it could have affected the thematic visualization.

This analysis only considers the literature published in nursing journals indexed in Scopus, which means that publications from other information resources or unpublished research or data have not been taken into account, and although, in this case, it has been possible to obtain an overview of what the scientific production on this topic was during the pandemic, the picture could be incomplete.

## 5. Conclusions

To the best of the authors' knowledge, this is the first study to analyze global publications on the different aspects related to death and dying amid the health chaos during the COVID-19 pandemic, and the results obtained offer a range of data and images. These data and images characterize the scientific production published on this particular topic, coming to the conclusion that, due to the multifaceted and multidisciplinary approach to the experience of death, care, and accompaniment in the dying process, bibliometric maps improve the comprehensive understanding of the semantic and conceptual structure of this field of research.

Although a specific period was taken as a reference for this research, in this case, the COVID-19 pandemic, due to the high mortality that occurred, these results can be generalized to any other period due to the relevance that the process of death and dying has for life itself.

Palliative and end-of-life care are priority issues in the scientific field. The results show the need to generate more knowledge about the best practices to provide dignified, humanized care centered on the patients' and their families, as well as to prevent and manage stress, suffering and grief for those involved in the process of dying, in addition to supporting the recommendation to increase international collaboration to facilitate the transformation of knowledge and practice in support of those countries with unmet needs

in the field of palliative care. Similarly, emerging research areas are identified, such as religion and spirituality or pediatric palliative care, which require further attention and development. Further studies addressing these issues from different perspectives and methodologies are necessary, and these studies should include the voices of patients, their families and health professionals, contributing to improving the quality of life and facilitating a more dignified death for people in a terminal situation.

**Supplementary Materials:** The following supporting information can be downloaded at: https://www.mdpi.com/article/10.3390/nursrep14020050/s1, File S1: Excel spreadsheet containing complete metadata of references retrieved for description and Analysis of Research on Death and Dying during the COVID-19 Pandemic, Published in Nursing Journals Indexed in SCOPUS. (References [31–149] are cited in the Supplementary Materials).

**Author Contributions:** Conceptualization, L.C.-P., C.-E.M.-A., M.M.N.-M. and J.Á.R.-G.; methodology, L.C.-P.; validation, N.R.-N. and Y.M.R.-N.; formal analysis, L.C.-P. and C.-E.M.-A.; investigation, L.C.-P.; data curation, N.R.-N. and Y.M.R.-N.; writing—original draft preparation, L.C.-P. and C.-E.M.-A.; writing—review and editing, N.R.-N. and Y.M.R.-N.; supervision, C.-E.M.-A., M.M.N.-M. and J.Á.R.-G. All authors have read and agreed to the published version of the manuscript.

**Funding:** This research received no external funding.

**Institutional Review Board Statement:** Not applicable. This study did not require ethical approval.

**Informed Consent Statement:** Not applicable.

**Data Availability Statement:** The study dataset is available through the following link: https://osf.io/hcz8f/ (accessed on 14 March 2024).

**Public Involvement Statement:** No public involvement in any aspect of this research.

**Guidelines and Standards Statement:** Despite the widespread use of bibliometric indicators in scientific analysis, there are no agreed-upon guidelines or standards for reporting bibliometric studies.

**Acknowledgments:** The authors would like to thank Patrick Dennis for his help in translating the manuscript.

**Conflicts of Interest:** The authors declare no conflicts of interest.

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
