# Peer review of "Description and Analysis of Research on Death and Dying during the COVID-19 Pandemic, Published in Nursing Journals Indexed in SCOPUS"

_nursrep, doi:10.3390/nursrep14020050_

Round 1

Reviewer 1 Report

Comments and Suggestions for Authors

Thank you for an interesting manuscript. 

I suggest to include more information about the used method in the method section. Many researchers and readers may not be familiar with the method. Therefore you should write more about the method and include references for those who might want to read more about it. 

Author Response

Dear Reviewer;

On behalf of all the authors and myself, I would like to thank you so much for taking the time to review our manuscript. We appreciate your suggestion about improving the methodology to promote the understanding of those researchers who are not familiar with bibliometrics.

In this sense, we have added a paragraph at the beginning of the methodology, in which we comment on the two large groups of analysis within bibliometrics and what can be analysed with each one. The four papers mentioned in this paragraph serve as excellent resources for learning more about this field, which is still challenging to comprehend and apply correctly even though it has been gaining traction outside of the information sciences, particularly in multidisciplinary and health sciences journals.

There are no standards for reporting about this kind of work, but there are rules to develop the analyses within quality limits. In our study, we have explained the methodology that we have developed through two of the most important techniques within this discipline: performance analysis through statistical-descriptive analysis with Bibliometrix and the technique of scientific mapping through labelling maps and density maps with VOSviewer. In this respect, we have described the indicators analysed, and we have also informed about the process of adjustment parameters and frequency thresholds until achieving good visualisation criteria, essential for getting optimal visualisation understanding of the VOSviewer representations. Likewise, we inform about the origin of the data as well as the reason for choosing the database, and we have added a detailed search strategy with the results of each row and the date the data was obtained. All these are essential details to guarantee the transparency and reproducibility of the bibliometrics studies.

In addition to the above, in the bibliography we have several references where it is possible to find information about the tools we have used for the analyses, as well as some other methodological issues and various bibliometric studies that can help to increase the understanding of bibliometry.

Once again we thank you for your comments and wish you a good day.

Kind regards,

Reviewer 2 Report

Comments and Suggestions for Authors

The article addresses an important issue from an interesting point of view. However, the current text fails to show its full potential. This needs a major revision to strengthen its impact and clarity.

- The introduction should be concise and focused, beginning directly with critically examining the existing literature in the main area of concern. Highlight the gaps and articulate the specific aim of the study, making it clear that the objective goes beyond the mere application of bibliometrics.

- Describe the methods concisely, including essential details while avoiding non-essential information. In the search strategy, explicitly state document selection criteria, the number and role of reviewers, and how they solved any discrepancies. Keep data extraction and analysis descriptions simple, reserving additional details for an appendix or references. Address ethical considerations briefly, even if covered later in the “Institutional Review Board Statement”.

Results: Select relevant information and consider using more engaging titles for the separate paragraphs, such as "main themes" or "main topics" instead of technical terms like "co-word analysis."

-Tables and figures: Reorganize tables for clarity (e.g., Table 1 from #1 to #5). Consider using diagrams to illustrate article inclusion and exclusion. Evaluate figures for relevance and potentially remove those deemed pedantic (e.g., Figure 1).

- Formulate a more straightforward discussion. Begin with a sentence restating the study's aim and methods. Discuss each relevant finding from the results, highlighting similarities and differences compared to existing literature. This will clarify the study's contribution to knowledge.

- Address any potential limitations from the methods used (e.g., reliance on a single search database, inclusion/exclusion criteria applied).

Conclusions: Here is the place for the implications for research and clinical practice (such as those provided in lines 407-419). Reconsider the need for the statement about a multidisciplinary approach (line 437), giving evidence before in the manuscript or omitting the statement if not supported by the study.

Minor: Correct punctuation and typos (e.g., line 89 Study desing). Ensure consistency in presenting authors ‘names (e.g., line 202 Betty Rolling Ferrell and William E. Rosa).

The manuscript holds a high potential; however, a significant text revision is essential. Focus on a clear message and organize the ideas logically and coherently to enhance accessibility to a broader audience.

Comments on the Quality of English Language

Correct punctuation and typos (e.g., line 89 Study desing). Ensure consistency in presenting authors ‘names (e.g., line 202 Betty Rolling Ferrell and William E. Rosa).

Author Response

Dear Reviewer;

On behalf of all the authors and myself, I would like to thank you so much for taking the time to review our manuscript. Your work has offered us the opportunity to take into account some formal issues that we could address differently, which will surely make our study better. Thank you, also, for thinking that our study addresses an important issue from an interesting point of view.

  • The introduction should be concise and focused, beginning directly with critically examining the existing literature in the main area of concern. Highlight the gaps and articulate the specific aim of the study, making it clear that the objective goes beyond the mere application of bibliometrics.

We have revised the introduction and deleted some paragraphs, retaining those where reference is made to some previous studies, in which different topics were addressed on issues that we consider to be of interest in relation to nurses during the pandemic (rows 52 to 60). We have also mentioned that the decision to conduct this research, responded to the fact that this topic does not appear to have been developed in previous studies and that, given the importance that have the different aspects within the process of death and death, and because the nurses care have a relevant role in this process, not only for patients but also for family members, seemed relevant to obtain a broad view of nursing research on the process of death and dying.

This broad vision implies not only the analysis of quantitative indicators but also to identify the main lines of research, those more developed and those that have shown a seemingly less interest, because of their scarce development. The oldest lines of research and the most current topics. The period of the pandemic was chosen for the significance of the high incidence of deaths worldwide.

  • Describe the methods concisely, including essential details while avoiding non-essential information. In the search strategy, explicitly state document selection criteria, the number and role of reviewers, and how they solved any discrepancies. Keep data extraction and analysis descriptions simple, reserving additional details for an appendix or references. Address ethical considerations briefly, even if covered later in the “Institutional Review Board Statement”.

The bibliometrics analyses, as a research methodology, it is different from a literature review, and although they may be complementary, they do not share design, objectives, techniques, or procedures. A quality bibliometric analysis does not require critical reading or peer review; there is no need to define selection criteria and therefore no need to resolve discrepancies between reviewers, not necessary to report the results of the search and selection through a flow chart as established in the PRISMA statement for reviews. The unit of analysis is also different. While reviews analyse the publications (original articles), bibliometrics is based on the bibliographic information contained in the records indexed in a database to identify patterns within a research group, a journal, country, research area, scientific discipline, etc.

The reports of methodology in the bibliometrics analysis also differ from reviews. The review reports are systematized because respond to a standardized methodology, however, although bibliometric has proven to be a valuable tool for understanding scientific activity and its impact, it is heterogeneous not only in terms of the different indicators that can or may not be analyzed depending on the aim of the researcher, but also the techniques and tools of analysis are diverse. At the end of the manuscript, in the Guidelines and Standards Statement section, we explained this particularity.

There are no standards for reporting about this kind of works, but there are rules to develop the analyzes within quality limits. In our study we have explained the methodology that we have developed, through two of the most important techniques within this discipline, I refer to performance analysis through statistical-descriptive analysis, with Bibliométrix and the technique of scientific mapping, through of the labelling maps and the density maps with VOSviewer. In this respect, we have described the indicators analysed, and we have also informed about the process of adjustment parameters and frequency thresholds until achieving good visualization criteria, to seeking a good understanding of the VOSviewer representations. Likewise, we inform about the origin of the data, as well as the reason for choosing the database and we have added the detailed search strategy, with the results of each row and the date the data was obtained. All these are essential details to guarantee the transparency and reproducibility of the bibliometrics studies.

Following your suggestion, we have added information on the ethical considerations of the study, in the Study Desing section, within the methodology.

  • Results: Select relevant information and consider using more engaging titles for the separate paragraphs, such as "main themes" or "main topics" instead of technical terms like "co-word analysis."

Done. The suggested title better identifies this section.

  • Tables and figures: Reorganize tables for clarity (e.g., Table 1 from #1 to #5). Consider using diagrams to illustrate article inclusion and exclusion. Evaluate figures for relevance and potentially remove those deemed pedantic (e.g., Figure 1).

Thank you very much. There was an error in the numbering of the tables, so this has been corrected. In addition, table 1, which was unnecessary, has been eliminated. Concerning the other point raised in this part, I must tell you that we did not establish selection criteria, nor did we include a flowchart such as that of the PRISMA statement, because this study did not analyze the articles, so all the references retrieved were included. However, as this is a limited corpus (only 119 papers), we reviewed the references during the process of designing the search strategy to check the relevance of the terms included. All results were eligible.

The point is that with this research methodology, the metadata contained in the references indexed in the databases is analysed instead of the articles, as is done in the reviews. The selection of bibliographic information is carried out during the data normalisation process. This is explained in the methodology.

  • Formulate a more straightforward discussion. Begin with a sentence restating the study's aim and methods. Discuss each relevant finding from the results, highlighting similarities and differences compared to existing literature. This will clarify the study's contribution to knowledge.

At the beginning of the discussion, we have introduced the information that you suggests to us about reaffirming the objective and methods of the study

  • Address any potential limitationsfrom the methods used (e.g., reliance on a single search database, inclusion/exclusion criteria applied).

Thank you very much. Among the limitations listed, the fact that only the Scopus database was used is already mentioned.

  • Conclusions: Here is the place for the implications for research and clinical practice (such as those provided in lines 407-419). Reconsider the need for the statement about a multidisciplinary approach (line 437), giving evidence before in the manuscript or omitting the statement if not supported by the study.

Among the results and discussion of this work, we talk about different lines of research located through the Vosviewer maps, which address the diversity and complexity of the needs and demands of terminal patients and their families, highlighting the need for comprehensive attention to the different facets confluent in the process of death and death, which includes emotional, ethical, spiritual, communicative, etc, for care at the end of life, grief, bereavement, loss, etc., which affect not only the patients but also their families and the health professionals. The multidisciplinary component identified through the lines of research described has to do with the need for medical and nursing care, psychological and religious care, etc. Even IKs were identified, related to the need for administrative and political regulation. All these issues point to the multidisciplinary approach mentioned in the conclusions.

Done. The implications for research and clinical practicewere including in this secction.

  • Minor: Correct punctuation and typos (e.g., line 89, Study desing). Ensure consistency in presenting authors ‘names (e.g., line 202, Betty Rolling Ferrell and William E. Rosa).

Thank you very much. We have corrected the authors' names and the section headings.

Reviewer 3 Report

Comments and Suggestions for Authors

Dear authors, first of all, I congratulate you on your work.

However, the article presents internal inconsistencies that must be improved to evaluate its publication.

First, the title discusses death and dying, but the keywords, discussion, and nursing implications focus on palliative care.

The stated objective is not measurable and is not answered by the type of study proposed, which must also be improved, as expressed in the comments.

Title: must be representative of the work performed.

Summary: The objective in the summary must be the same as stated throughout the text. Evaluate the relevance of the objective verb since this is the authors' objective with the publication, but not the very objective pursued by the development of this research. The method does not describe the eligibility criteria for critical reading of the articles for selection.

MeSH: Okay.

Introduction: delimits the object of study. However, it warrants further investigation. Likewise, it does not propose a specific objective from the investigative taxonomy; it provides an objective with a vague purpose.

It is requested that a concrete and measurable objective be proposed and a specific justification for this research, highlighting the social and scientific value to which it responds.

Methodology:

Type of study: The study to be developed must be clearly declared with reference to it. Supported by the justification given in the introduction.

Discussion: The lack of presentation of the articles stands out when making them talk about the results. The first time each article is used, an introduction should be made to raise the quality of the discussion.

Regarding limitations, the need to evaluate the quality of the articles is recognized, but how it could be implemented in this same type of research, is not mentioned.

Nursing implications: it stands out that they focus on palliative care since this is not the focus of the title or the objective of the research development. On the other hand, only two references that support this topic. Therefore, the references should be expanded in discussing this topic if it will be the focus of the Nursing implications.

Conclusions: These are difficult to evaluate due to the lack of coherence between the objective and type of study.

References: ok.

Comments on the Quality of English Language

Dear authors, first of all, I congratulate you on your work.

However, the article presents internal inconsistencies that must be improved to evaluate its publication.

First, the title discusses death and dying, but the keywords, discussion, and nursing implications focus on palliative care.

The stated objective is not measurable and is not answered by the type of study proposed, which must also be improved, as expressed in the comments.

Title: must be representative of the work performed.

Summary: The objective in the summary must be the same as stated throughout the text. Evaluate the relevance of the objective verb since this is the authors' objective with the publication, but not the very objective pursued by the development of this research. The method does not describe the eligibility criteria for critical reading of the articles for selection.

MeSH: Okay.

Introduction: delimits the object of study. However, it warrants further investigation. Likewise, it does not propose a specific objective from the investigative taxonomy; it provides an objective with a vague purpose.

It is requested that a concrete and measurable objective be proposed and a specific justification for this research, highlighting the social and scientific value to which it responds.

Methodology:

Type of study: The study to be developed must be clearly declared with reference to it. Supported by the justification given in the introduction.

Discussion: The lack of presentation of the articles stands out when making them talk about the results. The first time each article is used, an introduction should be made to raise the quality of the discussion.

Regarding limitations, the need to evaluate the quality of the articles is recognized, but how it could be implemented in this same type of research, is not mentioned.

Nursing implications: it stands out that they focus on palliative care since this is not the focus of the title or the objective of the research development. On the other hand, only two references that support this topic. Therefore, the references should be expanded in discussing this topic if it will be the focus of the Nursing implications.

Conclusions: These are difficult to evaluate due to the lack of coherence between the objective and type of study.

References: ok.

Author Response

Dear Reviewer;
On behalf of all the authors and myself, I would like to thank you so much for taking the time to review our manuscript. Your work has offered us the opportunity to take into account some formal issues that we could address differently, which will surely make our study better.

Having said this, I would like, with your permission, to answer some of the questions you have raised.
To begin with, I must say that bibliometrics is a descriptive and analytical science with a quantitative approach that studies the characteristics of scientific publications based on the bibliographic information indexed in databases, which makes it possible to analyse the development of the different areas of knowledge. It is an essential methodology for measuring scientific activity in terms of the productivity of authors, countries, institutions, etc., as well as offering patterns of consumption of scientific publications. It also makes it possible to define the main lines of research within an area of knowledge through the analysis of keywords, which is important for identifying the most developed topics, emerging topics, or those that have ceased to be of interest to the scientific community within that area of research. In this sense, it allows the identification of knowledge gaps that can be exploited by publishers as well as funding entities to promote and deepen research.

As a research methodology, it is different from a literature review, and although they may be complementary, they do not share design, objectives, techniques, or procedures. A quality bibliometric analysis does not require critical reading or peer review; there is no need to define selection criteria and therefore no need to resolve discrepancies between reviewers, nor is it necessary to report the results of the search and selection through a flow chart as established in the PRISMA statement for literature reviews. The unit of analysis is also different. While a review synthesises research through publications (original articles), bibliometrics is based on the bibliographic information contained in the records indexed in a database to identify patterns within a research group, a journal, country, research area, scientific discipline, etc.

Hereafter, we offer you our arguments in response to some of the points you made.

  • First, the title discusses death and dying, but the keywords, discussion, and nursing implications focus on palliative care

The concept of death and dying, as a process, implicitly encompasses other concepts such as grief, bereavement, mourning, "End of life care" "End of life decision-making" suicide, "Traumatic Death" death-related attitudes, "death education,"  etc. Our overall objective was to provide an overview of the research published in nursing journals in the Scopus database on the different aspects of death and dying, so we also included in the search strategy, all possible perspectives of analysis that including keywords such as thanatology OR religion OR spirituality OR philosophy OR anthropology OR sociology OR cultural OR ethical OR legal OR institutional OR "Life Span" OR "Legal Aspects" OR "Historical Perspectives" OR "Contemporary Perspectives" OR "Professional Issues". If you look at the search strategy, the term palliative care is not included, however, within the corpus analyzed it seems to be a prevalent line of research. In the title, we put "Description and analysis of research on death and dying..." instead of "Description and analysis of research on palliative care...", since our objective was not limited to palliative care.

  • The stated objective is not measurable and is not answered by the type of study proposed, which must also be improved, as expressed in the comments.

Our objective was to offer a global vision, which implies describing the panorama, in a general way, at an international level, of research on a very specific topic such as death and dying, in a very specific period 2020–2023, in a specific database such as Scopus, based on journals belonging to a specific area such as nursing. In short, we wanted to know what the nurses who published in these nursing journals were most interested in or concerned about, given the pandemic's high death toll. Who published the most, where the most research was conducted, how much research was conducted, how researchers interacted or collaborated, what the characteristics of this research were in terms of types of studies and publications, which nursing journals published on this topic, and so on. All this information provides important clues for a comprehensive analysis, and it would be very difficult to describe these aspects using another research methodology. To analyse and describe these issues, we used statistical-descriptive analysis and the Bibliometrix programme, as we established in the specific objectives.

On the other hand, to fulfill the general objective of offering a global vision on this topic, we wanted to know what were the main concepts and focuses of interest in research on death and dying published in these journals. This can be done through a Scoping Review, but to be consistent with the selected methodology, we developed this task through the analysis of the keywords contained in the bibliographic records of the published articles, which helped us to delimit the different lines of the research described in the article. To better understand this aspect of the bibliometric analysis, we used the co-word map visualisation technique provided by the VOSviewer programme, as described in the previous specific objective.

  • Discussion: The lack of presentation of the articles stands out when making them talk about the results. The first time each article is used, an introduction should be made to raise the quality of the discussion

The unit of analysis in bibliometric research is the bibliographic information of the publications. We speak of metadata (author, title, date, keywords, etc.), not of the articles. This is an essential difference between a bibliometric analysis and a review of the scientific literature. The discourse is therefore different in each of these 2 research methodologies.

  • Regarding limitations, the need to evaluate the quality of the articles is recognized, but how it could be implemented in this same type of research, is not mentioned.

In the limitations, we do not say that it is necessary to evaluate the quality of the articles, but that bibliometric analyses do not take into account the quality of the published works, since the objective is not to evaluate the scientific evidence but to analyse and describe the publication patterns. In this sense, for example, no type of publication (letters, editorials, etc.) is discarded, since this information is an indicator of quality. Another pattern of quality could be the impact factor of the journals in which the analyzed papers are published, among others. This is the type of qualitative information that can be obtained through a bibliometric analysis.  These data are described in our study.

  • Nursing implications: it stands out that they focus on palliative care since this is not the focus of the title or the objective of the research development. On the other hand, only two references that support this topic. Therefore, the references should be expanded in discussing this topic if it will be the focus of the Nursing implications.

3; 4; 6; 7; 8; 9; 10; 11; 13; 15; 16; 17; 19; 32; 34; 36; 39; 41; 44; 47. All these references pertain to publications on nurses' care during the COVID-19 pandemic, publications on Covid-19 in nursing journals, nursing research on Covid-19, reflections on nursing in the context of COVID-19, etc. The rest of the references have served as a supporting argument regarding the analysis techniques and bibliometric programs we have used and also as confirmation of some of the results we have discussed in this work since they are results comparable to ours, which attests that the patterns and trends in research are universal and generalizable so they can serve as a reference point to explain the phenomena or findings in any area of knowledge, providing veracity and consistency to new studies.